# Influence of Additives on the Mechanical Characteristics of Hardox 450 Steel Welds

**DOI:** 10.3390/ma16165593

**Published:** 2023-08-12

**Authors:** Saulius Baskutis, Jolanta Baskutiene, Egidijus Dragašius, Lina Kavaliauskiene, Neringa Keršiene, Yaroslav Kusyi, Vadym Stupnytskyy

**Affiliations:** 1Department of Production Engineering, Faculty of Mechanical Engineering and Design, Kaunas University of Technology, Studentu St. 56, 51424 Kaunas, Lithuania; jbask@ktu.lt (J.B.); egidijus.dragasius@ktu.lt (E.D.); lina.kavaliauskiene@ktu.lt (L.K.); 2Department of Mechanical Engineering, Faculty of Mechanical Engineering and Design, Kaunas University of Technology, Studentu St. 56, 51424 Kaunas, Lithuania; neringa.kersiene@ktu.lt; 3Department of Engineering Mechanics and Transport, Lviv Polytechnic National University, Bandera St. 12, 79013 Lviv, Ukraine; jarkym@ukr.net; 4Department of Robotics and Integrated Mechanical Engineering Technologies, Lviv Polytechnic National University, Bandera St. 12, 79013 Lviv, Ukraine; vadym.v.stupnytskyi@lpnu.ua

**Keywords:** welding, Hardox 450 steel, mechanical characteristics, additives, weld zone

## Abstract

The aim is to overcome the issues of high-hardness material welding by different additives used to achieve the desired improvements. The research is focused on Hardox 450 steel welding and factors to be considered in order to maintain the required mechanical properties of the weld. The selection of best suited welding materials or additives, including filler metals and shielding gases, are within the important factors to be taken into account. During the welding of Hardox 450 steel, cobalt, nickel, tungsten and titanium additives and cobalt and tungsten mixture additives were used and their influence on the microstructure and mechanical properties of the fusion and heat-affected zones was investigated. The microstructure of the weld zone is related to certain mechanical properties of the weld and heat-affected zone, such as hardness, tensile and bending strength, yield strength, strain at ultimate tensile strength, the Young’s modulus and elongation. Research has shown significant differences in the mentioned parameters depending on specific additives used in the welds. It can be concluded that tungsten, used as an additive, increased the hardness of the heat-affected and fusion zones up to 478 HV; the combined presence of cobalt and tungsten additives improves the strength of the seam up to 744 MPa during tensile; and in the case of bending, nickel, when used as an additive, increased ductility (the bending modulus reached the limit of 94 GPa) and at the same time, decreased the risk of cracking. The obtained results highlight the possibilities for strengthening the welded joint of Hardox 450 steel using different additives or their mixtures. The research conclusions and recommendations aim at improving the quality and mechanical properties of welded Hardox 450 steel joints in various applications.

## 1. Introduction

High-strength steels are widely used for protective elements in transportation, military and mining industries. Armor, bunkers, containers, dump truck bodies and other machine parts are made of these steels. The parts must have excellent properties such as high strength and hardness, impact and abrasive-wear resistance, high welding performance, cold forming and fatigue resistance [1,2]. Hardox steels perfectly meet these needs. Furthermore, the parts made of Hardox steels are characterized by high durability, even when used in harsh environments [3]. Hardox steels have a particularly high impact resistance not only at room temperature, but also at relatively high and low temperatures [4]. It is important to note that the boronizing (boride layer) of Hardox 450 steel significantly increases its shielding properties against the effects of X-rays and gamma ray radiation, which is rather relevant in the production of protective armor in the military industry [5]. It is worth mentioning that a rather important criterion from the point of view of the practical application of Hardox 450 steel is its weldability [6,7]. Also, due to the relatively low carbon equivalent, all Hardox grades have a better resistance to hydrogen cracking if compared to other wear-resistant steels. However, Hardox is a relatively low-alloy martensitic steel, is sensitive to thermal loads, and therefore, Hardox 450 welding often becomes difficult and problematic. Regardless of the fact that relatively high mechanical properties of welded joints are obtained, due to unavoidable thermal processes during welding, undesirable changes in microstructures occur in areas affected by high temperatures.

In order to ensure the quality of welds, it is important to pay attention to factors such as crack morphology, the nature of crack propagation, the formation of phases and the mechanism of cracking in the weld [8]. Fang et al. [9] analyzed the substructure of inhomogeneous alloys, in particular, the influence of sub-grain boundaries and segregation of intragranular cellular elements on microstructure evolution and mechanical behavior of samples during the tensile process.

Excessive heat during welding increases the width of the heat-affected zone (HAZ), which in turn degrades the mechanical properties of the HAZ. This negatively affects significant changes in hardness, local loss of wear resistance and reduced plasticity [10]. Welding with a relatively low amount of heat increases the wear resistance of the HAZ, reduces distortion and increases the toughness and strength of the welded joint. Although an increase in the austenitizing temperature does not significantly affect the decrease in tensile strength, it has been shown that the austenitizing temperature is a critical parameter that determines the impact properties of Hardox 450 [11]. A very small amount of heat input to the weld zone can negatively affect the impact toughness. The mentioned changes are related not only to the welding parameters, but also to the additional materials used in welding and the mechanical and thermal processing operations of constructional elements made of Hardox steels [12].

In Gupta et al. [13], the research of Hardox 400 steel welding using ferritic filler electrodes due to the addition of higher alloying content in terms of Cr, Ni, Mn and Mo in the fusion zone (FZ), more refined and homogeneously distributed microstructural constituents were observed.

Conducted research [14,15,16] revealed that cobalt increases strength and wear resistance, fines the structure, reduces the amount of carbon in pearlite, accelerates the decomposition of austenite and reduces thermal conductivity. Since cobalt has a relatively negative effect on hardenability, it affects the hardening less than other additives.

The use of tungsten as an additional material during welding increases hardness, particularly at high temperatures, and heat resistance, promotes the formation of a fine homogeneous structure, reduces the amount of carbon in pearlite, stabilizes ferrite and promotes the removal of carbon from the surface of the metal during heating. In addition, tungsten increases the yield strength and tensile strength of steel without adversely affecting ductility and fracture toughness [17,18,19]. Adding tungsten together with cerium to the filler metal improves the high temperature creep strength of the weld joint [20]. In addition, it was found that the use of cerium in the weld not only improves the cryogenic impact toughness, but also increases the yield strength, tensile strength and total elongation of the weld [21]. Nickel as an additive increases the impact strength, plasticity, corrosion resistance and slight hardness, and promotes the formation of a fine structure. Nickel not only reduces or maintains a constant impact strength, but also reduces the brittleness of the metal in the cold [22,23,24]. In addition, the use of nickel as a filler is a reliable way to prevent cracking in laser-welded NiTi/stainless steel joints [25]. The combined presence of nickel and molybdenum in the weld metal allows the formation of a high volume fraction of fine acicular ferrite with good toughness [26]. The addition of molybdenum to the weld zone also results in the precipitation of Mo- and Cr-rich carbide at the grain boundaries of the δ-ferrite phase and influences compositional changes at the base metal/weld metal interface and in the FZ. The formation of these precipitates causes increased the hardness and strength of the weld [27]. Manganese and chromium tend to increase the hardness of steel. When 5% or more chromium is used in conjunction with manganese, the critical quench rate is reduced to the point where the steel is air-hardened. Chromium can also increase the wear resistance of steel. One of the most well-known effects of chromium on steel is its resistance to staining and corrosion [28,29,30,31]. Because titanium oxidizes easily, care must be taken when adding it as an additive. The strengthening mechanism of a high-strength low-alloy (Hardox) steel depends on a combination of carbide and nitride precipitation and grain refinement. Here, titanium can be used as a micro-alloying element, especially for the precipitation of titanium carbonitride (TiCN) after rapid cooling [32,33,34]. TiCN is the only micro-alloyed carbonitride that is stable at the high temperatures reached in HAZ welding where it reduces grain growth and increases toughness [35]. It should also be noted that results of the fatigue tests of welded joints showed that the weld metal microstructure without titanium additives was better resistant to fatigue crack growth [36]. The addition of a boron additive during welding reduces the grain boundary ferrite and ductile-brittle transition temperature, thus improving the impact toughness of the joint [37]. Studies have shown that adding even a small amount (about 0.01%) of boron combined with nitrogen suppresses the fracture in HAZ and improves the long-term creep rupture strength of welded joints [38]. It was found that adding a small amount of zirconium to the molten pool eliminated the porosity defects in the weld zone and increased the ultimate tensile strength of the joints from about 65% compared to the base metal [39]. Under the influence of zirconium as an additive, due to the formation of intragranular ferrite, the impact toughness of the HAZ is improved, especially when a large welding heat input is used [40]. The use of ruthenium as an additive in the welding of stainless steel 316L allows for the increase of the hardness of the button welds [41].

Although Hardox 450 steel has good weldability, if compared to some other steels, the negative structural changes in the material caused by the welding process in the welding zone deteriorate the mechanical properties of the welding zone. Experimental studies showed that the mechanical properties of the welding seam can be improved, and the negative structural changes caused by the welding process can be reduced by inserting additional materials into the welding zone.

This study aimed to evaluate the use of welding additives to prevent or reduce negative changes in the weld zone caused by welding and to determine the macro and micro structural changes of the weld seam in order to improve the mechanical properties of the Hardox 450 steel joints. The welding parameters determined during the research as well as the corresponding additives used for Hardox 450 steel joining made it possible to achieve the necessary strength characteristics of the weld and work on further developments and practical uses in the industry. Some of the obtained results serve as recommendations for Hardox 450 steel welding using additives and their mixtures.

## 2. Materials and Experimental Procedure

### 2.1. Materials and Equipment for Welding

Welded joints of Hardox 450 steel were made by tungsten inert gas (TIG) with the use of the Sherman DIGITIG 200AC/DC Multipro welding machine, tungsten electrode WL 20 (Ø2.4 mm), filler rods ER 70S-6 and E 7018. WL 20 electrodes are primarily composed of tungsten and contain 2% of thorium oxide. Thorium oxide is added to improve the electrode’s performance and stability. The welding rod is covered with a thin and homogeneous copper coating, which increases the resistance to rusting. In order to protect the weld pool from the influence of the atmosphere, 100% argon was used as the shielding gas, which provides a wide, shallow penetration weld bead and allows the arc length to be varied without disturbing the heat of the arc. The material being welded was sheets of 6 mm of thickness of the low-alloy high-strength steel Hardox 450. Table 1 and Table 2 show the manufacturer-declared properties of the Hardox 450 and the filler wire rods.

### 2.2. Preparation of the Specimens before and after Welding 

Sheets for investigation were prepared according to the recommendations specified in EN ISO 9692-1:2013 [42] for the single V-butt weld with a broad root face (Figure 1).

The groove angle α of 60° was formed on a milling machine. The gap (b) between two weldments was maintained at 2 mm. Before the welding procedure, all prepared samples were degreased with ethanol and polished to remove surface impurities, dusts and oxides.

The welding conditions of the specimens are summarized in Table 3.

All the welded sheets were classified into nine groups depending on the additives and welding wire used (Table 4). The checkmark (✓) in Table 4 indicates which additives and filer rods were used for Hardox 450 steel specimens welding.

Titanium, tungsten, nickel and cobalt powders of a 100~130 µm particle size range were used in this experiment. The powder was poured by evenly distributing it over the entire length of the V-groove (240 mm) after the first root pass. The weld joints were completed in four passes. Figure 2 shows hardness, metallographic studies, tensile and bending specimens’ distributions of the Hardox 450 welded sheets.

After completion of welding, all welded sheets were cleaned again and cut at a direction perpendicular to the weld for transverse tensile, bend and hardness tests and for a microstructure analysis of the weld zones. The cutting of the samples into the test pieces was done using a Bystronic Bysprint 3015 4 kW (Bystronic Laser AG, Niederönz, Switzerland) fiber CNC laser cutting machine. There were six specimens in each group for the tensile and bending experiments.

It is important to mention the fact that the welded sheets of group 9 were not further used for experimental studies, because the seam fractured due to the incompatibility of the materials in the welding pool and the resulting residual stress (Figure 3).

### 2.3. Mechanical Testing

The experiments and follow-up studies are summarized in Figure 4.

Hardness measurements along the welded joints (base metal-HAZ-seam-HAZ-base metal) were accomplished according to EN ISO 9015-2:2016 [43] by the Mitutoyo Hardness Testing Machine HM-210D (Mitutoyo Corporation, Kawasaki, Japan) using a diamond indenter under the load of 5.0 N, with 10 s of dwell time at 0.70 mm intervals, maintaining the rows of the indentations’ distance to the reference line (surface or fusion line) equal to 2 mm, until the original hardness of the base metal (BM) was reached.

Tensile and bend tests were performed, aiming to evaluate the mechanical properties of the weld. The tensile tests were performed using a 100 kN versatile electromechanical testing machine with Hottinger Baldwin Messtechnik GmbH (HBM) testing devices: force transducer U5 (HBM), accuracy class 0.1, nominal force 100 kN, at the lowest measurable value of 6 N; strain sensor DD1 (HBM), nominal displacement ±2.5 mm, accuracy class 0.1; displacement transducer WA-50 mm (HBM), accuracy class 0.2, measurement range 50 ± 0.1 mm; and measuring amplifier Spider8 (HBM). Tensile specimens were prepared perpendicular to the weld seam. The specimens were deformed at a crosshead speed of 10 mm/min until their complete failure. The welded joint of the specimen was not subjected to heat treatment in accordance with the regulations for the welded joint to be tested. The specimens for tensile tests were prepared according to the EN ISO 4136:2012 [44] standard as presented in Figure 5a,b.

Bend tests of the welded specimens were carried out using universal tensile test machine Tinius Olsen H10KT (Tinius Olsen Ltd., Salfords, Redhill, UK), load capacity 10 kN, at a room temperature of 22 ± 1 °C, with 50 ± 5% relative humidity, at a test speed of 2 mm/min and at a distance of 35 mm between rollers, with a force measuring sensor 10 kN and accuracy class of 0.1.

### 2.4. Metallographic Preparation

Specimens for optical examination were prepared using metallographic preparations (Figure 6).

The cut specimens were placed in a synthetic resin mixture, ground and polished. Etching of the polished surfaces was conducted using a spirit solution of 3% nitric acid (HNO_3_).

An optical investigation of the welded joints was carried out with an optical metallographic microscope Carl Zeiss Scope A1 (Carl Zeiss Microscopy GmbH, Jena, Germany).

## 3. Experimental Results 

### 3.1. Microstructure

Hardox 450 steel obtained by thermomechanical rolling, from which the specimens were made, has a characteristic microstructure of quenched martensite, the morphology of which is similar to slate with areas of tempered martensite. The formation of such a microstructure is influenced by self-tempering processes after quenching treatment carried out under metallurgical conditions immediately after rolling. The microstructures of FZ, HAZ and BM of specimen groups 1–8 are presented in Figure 7, Figure 8, Figure 9, Figure 10, Figure 11 and Figure 12.

As shown by studies of weld seam structures, in order to avoid unmelted additive particles, it is important to review and adjust welding parameters such as the welding current, voltage and travel speed to ensure adequate heat input and additive metal melting. Figure 13 shows particles of unmelted nickel due to improperly selected welding modes.

### 3.2. Tensile Test

The tensile strength test is one of the commonly performed tests to analyze the weldability of materials and the reliability of welds. The aim of the static tension transverse weld seam test was to determine the main mechanical strength characteristics of the seam of the HARDOX 450 welded specimens—stresses and deformations in the weld zone and the location of the fracture in the specimen.

According to the EN ISO 6892-1:2016 standard [45], the values measured and recalculated during the transverse tensile tests are presented in Figure 14 and Figure 15 and in Table 5: the ultimate tensile strength, yield strength, Young’s modulus and strain at ultimate tensile strength. The value of the elastic modulus during the transverse tensile test was determined at 10–40% of the conventional yield strength.

After the sample fracture during the experiment, the fractured surfaces and their nature were investigated. The registered defects that could negatively affect the test results, a comparison of the elastic modulus and the conventional yield strength, ultimate strength and strain at the ultimate strength of the HARDOX 450 samples determined during the transverse tensile test are presented.

During the test of static tension across the weld seam, Hardox 450 samples of groups 1–2 and 5’s fractured ductile, and samples of groups 1–4 and 6–8 showed a brittle fracture (Figure 16). The place of fracture of the specimens of all groups is through the seam.

### 3.3. Bend Test

In order to evaluate the plasticity of the material of the welding zone and possible surface defects’ influence on the specimen fracture, the bend tests of Hardox 450 specimens with welding seams were performed. The experimental setup for bend tests is presented in Figure 17a. The applied standard-EN ISO 5173:2010/A1:2013 [46] specifies the specimen sizes and the procedure for performing bend tests in order to determine the bending modulus and yield strength of the butt weld of the transverse profile bending specimen, bend angle and percentage elongation after failure in bending.

When the specimen is bent elastically, the bending modulus of the material during bending is determined by measuring the applied load and deflection. Based on the magnitude of the force determined during the bend test, the bending yield strength was calculated. The value of the bending modulus in bending is set at 10–40% of the conventional yield strength. During the bending of the specimen, at the point of the maximum bending moment, when the stress reaches the yield point, a so-called plastic hinge is formed and the specimen bends but does not break, so it is not possible to determine the strength limit of Hardox 450 steel welded samples during bending. The bend angle *α* (degrees) of the specimen under the force was calculated based on the displacement of the former according to the standard EN ISO 7438:2020, annex A [47]. After the bend tests, the fracture surfaces of the welded seams were visually examined (Table 6), and the results of deficiencies/defects that could negatively affect the test were recorded. A comparison of the average values of the bending modulus and yield strength in bending (Table 6), bend angles and percentage elongations after fracture (Figure 18) of Hardox 450 steel specimens welded with different additives are presented.

### 3.4. Hardness Test

The hardness profiles across the weld joints’ interface are presented in Figure 19. The hardness measurement experiments showed that the welding process led to a wide range of structural changes in Hardox 450, resulting in relatively large areas of reduced hardness (Figure 19; 1 and 2 specimen group). These changes in hardness reduction in the HAZ and FZ are determined by the applied welding technology, welding parameters that determine heat input, used base material and the filler rod, and accordingly, on the initial microstructure of the welded steel. Typically, Hardox 450 steel is delivered from the factory in a quenched and tempered condition [6]. This process results in a martensitic steel microstructure, which contributes to its relatively high hardness.

## 4. Discussion

### 4.1. Microstructure Characterization

In Hardox 450 steel welds, the FZ usually has a dendritic microstructure (Figure 7b,d). Dendrites are crystalline structures formed during solidification. The size and shape of the dendrites largely depends on the welding parameters and the cooling rate. In the region of HAZ closest to the FZ, also known as the excessive heating zone, a coarser microstructure is formed if compared to the BM (Figure 7a,c). The formation of the coarse-grained structure is influenced by the hardening processes taking place at high temperatures and the relatively low carbon content in this zone of the welded joint. Grain growth in this region can lead to reduced toughness and increased crack susceptibility. The middle region of the HAZ, known as the annealed zone, is characterized by a finer microstructure with better strength if compared to the excessive heating zone. In the HAZ region adjacent to the unaffected BM, known as the inter-critical HAZ, a partial austenite transformation occurs. Most often, the microstructure in this region consists of a mixture of ferrite and austenite. The presence of austenite in this region reduces the hardness, which was observed during the hardness measurement (Figure 19; 1 and 2 specimen groups).

Microstructure studies have shown that the use of tungsten and cobalt as welding additives affect the formation and properties of the HAZ and FZ microstructure. Both additives have a strong grain-refining effect, which leads to the formation of smaller grains in the HAZ and FZ (Figure 8a,b and Figure 11a,b). As already mentioned, Hardox 450 steel usually has a certain amount of retained austenite in its microstructure. Additives of tungsten and cobalt influence the stabilization of retained austenite, which helps prevent its transformation into martensite during welding. As experimental studies have shown, this has a positive effect on the strength and flexibility of the welded joint.

The use of tungsten as an additive in the welding process showed a positive effect on the structure formation of Hardox 450 steel welds. Tungsten reacts with carbon to form tungsten carbides during the welding process (Figure 9a,b) [48]. The carbides formed are extremely hard and contribute to the overall hardness and wear resistance of the weld. This statement was confirmed by hardness measurements (Figure 19; 4 specimen group).

Nickel as an additive in welding has a strong austenite stabilization, which means that it promotes the formation of the austenite phase in Hardox 450 steel (Figure 10a,b) [49]. During welding, the addition of nickel helps maintain the austenitic microstructure of the weld metal even during rapid cooling. Austenite has a face-centered cubic crystal structure that exhibits high ductility, making it desirable for welding [50]. In addition, nickel as an additive in the weld improves the corrosion resistance of the Hardox 450 steel weld.

Cobalt has a stabilizing effect on retained austenite [51], which is a metastable phase formed during the welding (Figure 12a–d). Retained austenite has good toughness and ductility. Although the addition of cobalt does not express a specific mechanical property so strongly, its addition helps to maintain the stability of the retained austenite and improve the general mechanical properties of the weld, especially at higher temperatures.

### 4.2. Mechanical Properties

As experimental studies have shown, tungsten affects the tensile strength of the weld, increasing the hardenability of steel and solid solution strengthening [52]. It forms a solid solution with iron, which increases the strength, hardness and slight plasticity of the material. This is mainly attributed to the increased strain hardening rate resulting from the already mentioned tungsten solid solution [53]. This conclusion is confirmed by the obtained results of hardness measurements (Figure 19; 4 specimen group) and tensile tests (Figure 15). An interesting fact was observed when tungsten and cobalt additives were placed in equal parts in the welding zone. Their overall strengthening effect significantly increased the tensile strength of the welds, as these additives influenced solid solution strengthening, grain refinement and the presence of hard particles, which improved mechanical properties, making the welds stronger and more resistant to deformation and cracking. In addition, cobalt as an additive also helps prevent the formation of brittle phases such as martensite, which reduces the ductility and toughness of the welds [54]. Tests on welded specimens did not show that the addition of nickel to the welds of Hardox 450 steel would have a significant positive effect on the strength of the weld in the tensile test. The highest average values of the ultimate strength during tensile tests were reached in the case of specimen groups 6 and 3, respectively, 774 MPa and 760 MPa, which contained tungsten and cobalt additives (Figure 14 and Figure 15).

In a bend test, both tensile and compressive stresses occur in the specimens.

The nickel as an additive during welding improves the weldability of Hardox 450 steel. Nickel is known as an additive which increases the ductility of steel and is important in reducing the risk of cracking during welding. Nickel acts as a solid solution enhancer and can increase resistance to hydrogen-induced cracking [55]. This undesirable phenomenon most commonly occurs when welding high-hardness steel grades such as Hardox [56]. In addition, nickel influences the formation of tough microstructures in the HAZ of the weld, which is adjacent to the FZ where thermal cycling occurs during welding. These tough microstructures increase the overall ductility and resistance to the brittle fracturing of the welded joint.

Experimental studies showed that the tungsten additive has no tangible direct influence on the mechanical properties of Hardox 450 steel welds during the bend test. The addition of tungsten increases the hardness (Figure 19; 4 specimen group) and strength (Figure 17b and Figure 18 and Table 6) of the weld, reducing its ductility and elongation during bending. Tungsten as an additive led to a more brittle behavior of the weld, so the weld is less plastically deformed without fracturing. This fact can affect the ability of the weld to withstand bending forces and lead to an increased risk of cracking or failure under bending loads in the welded structure. Similar to the case of tungsten, the cobalt additive’s influence on the behavior of welds during bend tests can also be characterized. A higher cobalt content can increase the brittleness of the weld and, at the same time, reduce the plastic deformation.

The visual inspection of the bending specimens revealed that during static bending, the bending HARDOX 450 steel specimens of groups 1–2 and 5 were ductile fractured. Opened cracks are visible at the edge of the tensile zone of specimens of groups 1 and 2, where the highest tensile stresses occur during bending. In specimen groups 3, 4, 6, 7 and 8, a brittle fracture was recorded in the tensile zone of the specimens. Initial defects leading to bending failure are visible as additional stress concentrators. No cracks were detected outside of the welding seam of group 5’s specimens after the bend test. Experimental bending tests showed that the stresses in the cross-sections of the specimens are distributed non-linearly during bending, because the yield point is first reached in the outer layers of the specimen, and cracks appear in the seams, except for the specimens of group 5, where nickel as an additive increased the ductility of the seam (Table 6). The highest average values of specimen bend angles α, calculated on the basis on the displacement of the former during the bending, were reached in specimen groups 1 and 2 (≈60°). The smallest bend angles of 30 and 33 ° were obtained in specimen groups 3 and 4 (Figure 18).

In the case of welding, the decrease in hardness compared to the BM is attributed to the tempering processes taking place in the HAZ and FZ.

When tungsten powder is included as an additive to the welding seam, a significant increase in hardness of the FZ is observed (Figure 19; 4 specimen group). Tungsten forms hard carbides with carbon in the weld pool. The tungsten additive can act as a steel grain refiner [57]. It promotes the formation of fine and uniform grains during the solidification process, which improves the mechanical properties of the weld, such as hardness. The weld containing tungsten can be heat-treated to achieve a high level of hardness and strength throughout the weld cross-section. This is particularly useful for welding components that require a high surface hardness and core strength. Experimental studies have shown that the addition of cobalt as an additive along with tungsten slightly decreases the weld hardness (Figure 19; 3 and 6 specimen groups). Cobalt stabilizes retained austenite, whereas tungsten can contribute to its formation. The combination of cobalt and tungsten as additives in Hardox 450 steel welds can result in a higher volume fraction of retained austenite. This improves the ductility of the weld metal with a reduction in hardness. The use of the nickel additive in welding Hardox 450 steel has a relatively small effect on the hardness of FZ and HAZ (Figure 19; 5 specimen group). Nickel is more known for its excellent corrosion resistance properties [58]. The addition of nickel to the weld increases the corrosion resistance of Hardox 450 steel. It helps to form a protective oxide layer that acts as a barrier against corrosion and improves the weld seam resistance to various corrosive environments. The use of the cobalt additive in Hardox 450 steel welding relatively increases the hardness slightly (Figure 19; 7 and 8 specimen groups), whereas the formation of cobalt containing carbides in the weld gives the weld better resistance to abrasion and wear.

## 5. Conclusions

The main conclusions have been drawn based on the research results.

Experimental studies of hardness measurements showed that when welding Hardox 450 steel, a decrease in hardness is observed in the FZ and HAZ, if compared to the BM. This is related to the tempering processes that take place in the HAZ and FZ. In order to increase the hardness of the mentioned zones to at least the BM, it is recommended to use appropriate additives during welding of Hardox 450 steel, one of which would be tungsten.

After performing a comparison of the average values of the calculated yield strength during tension, it was found that the highest values were reached in the case of specimen groups 6 and 3 after adding equal parts of tungsten and cobalt powder as additives to the weld, 586 MPa and 678 MPa, respectively. In the case of group 5 of the specimens, using the nickel powder additive, the highest average value of strain (4.63%) at the ultimate tensile strength was determined. The highest average value of Young’s modulus during tension was determined in welded specimens of group 6 and reached 235 GPa.

The highest average values of the bending modulus of the specimens during bending were determined in the specimens of groups 1 and 5. In the case of group 5 specimens, when using nickel powder as an additive in the weld, the bending modulus values reached as high as 94 GPa. A comparison of the average values of the calculated yield strength during bending showed that in the case of specimen groups 3 and 6, they reached the highest values of 492 MPa and 507 MPa, respectively. The highest average values of percentage elongations after fracturing were determined for groups 1–2 and 5 of the specimens—233% and 296%, respectively. The lowest elongation was determined for the specimens of group 4 (123%) using tungsten as an additive.

Experimental studies have shown that an excessive amount of titanium promotes the precipitation of large particles in the steel matrix, which greatly deteriorates the mechanical properties of the welded joint.

Tests on the microstructures of the specimens showed that the tungsten and cobalt additives used together have a strong grain-refining effect, resulting in the formation of finer grains in the HAZ and FZ. These additives influence the stabilization of retained austenite, which helps to prevent its transformation into martensite during welding, and at the same time, has a positive effect on the strength and flexibility of the welded joint. During the welding, the tungsten additive reacts with carbon, resulting in the formation of tungsten carbides, which are extremely hard and increases the hardness and wear resistance of the weld. The nickel additive used during the welding helps to maintain the austenitic microstructure of the weld metal and ensures the weld’s joint ductility.

## Figures and Tables

**Figure 1 materials-16-05593-f001:**
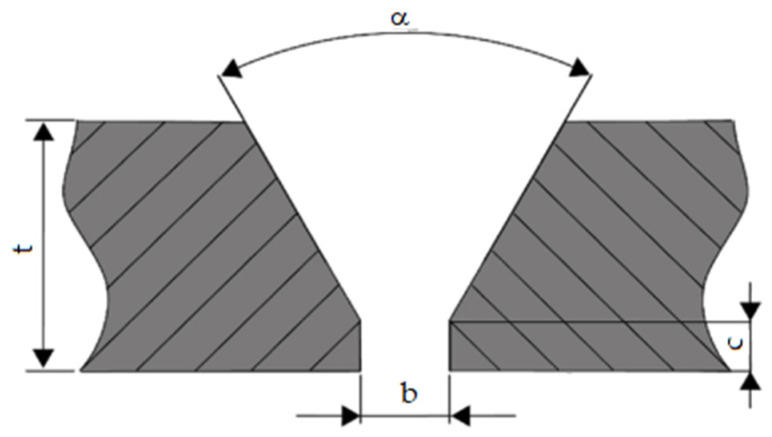
Cross section of sheet edge: α—an angle of the groove edge; t—the thickness of the weldment; b—root opening, c—thickness of root face (1.5 mm).

**Figure 2 materials-16-05593-f002:**
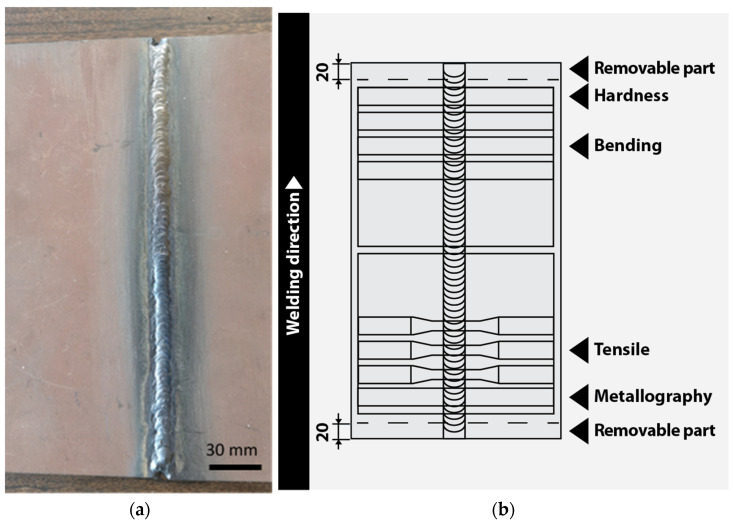
Welded sheets of Hardox 450 steel: (**a**) Sheets for laser cutting; (**b**) Sketch of sampling of welded sheets.

**Figure 3 materials-16-05593-f003:**
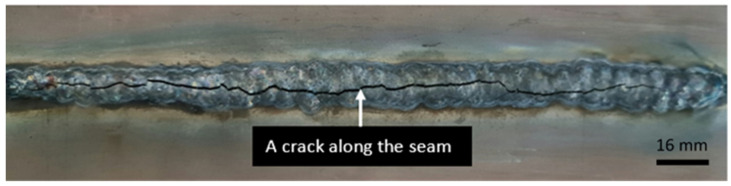
The weld seam was damaged by the defect formed after welding the sheets of group 9.

**Figure 4 materials-16-05593-f004:**
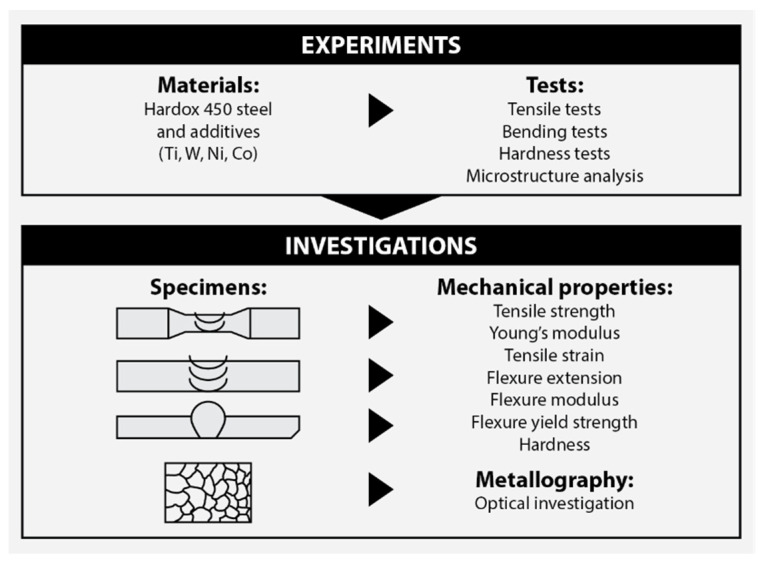
Content of experimental studies and analyzed parameters.

**Figure 5 materials-16-05593-f005:**
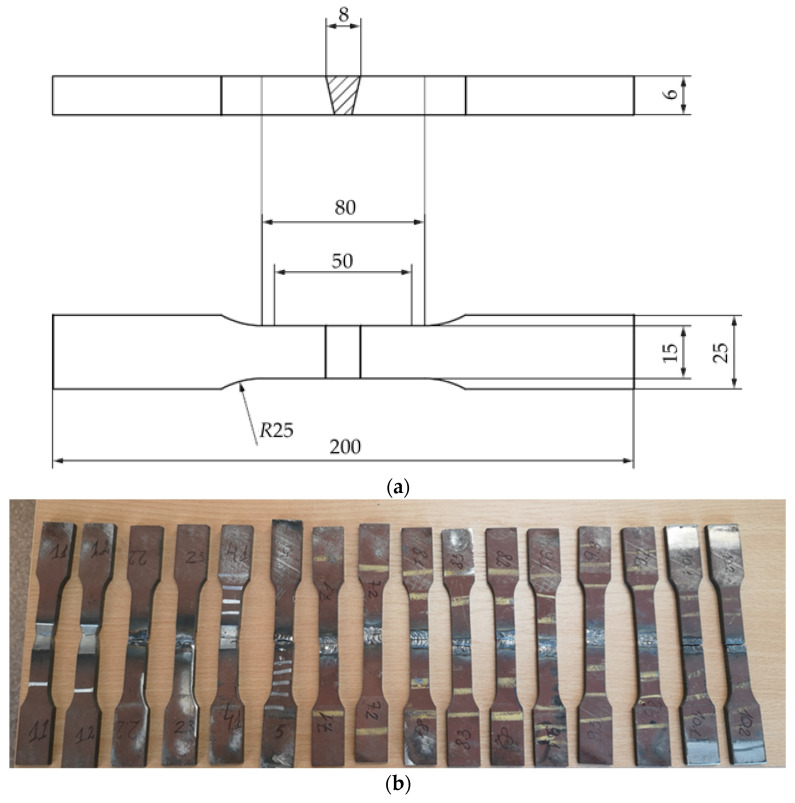
Tensile specimens: (**a**) Tensile specimen geometry; (**b**) Laser cut specimens prepared for tensile testing.

**Figure 6 materials-16-05593-f006:**
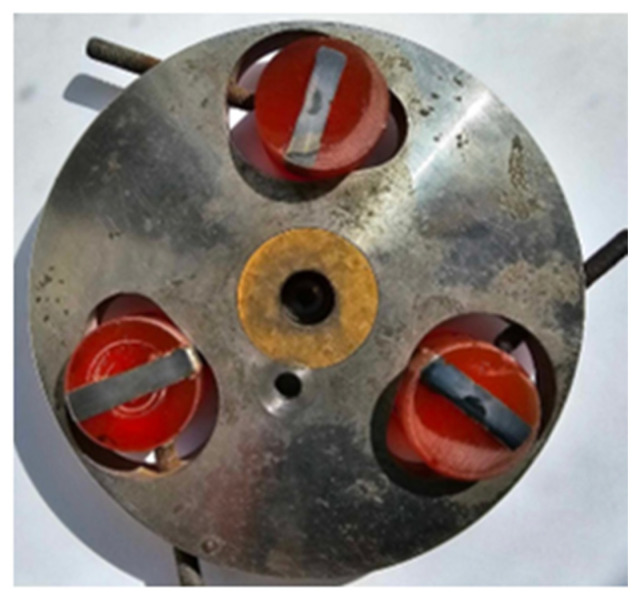
Specimens for optical investigation.

**Figure 7 materials-16-05593-f007:**
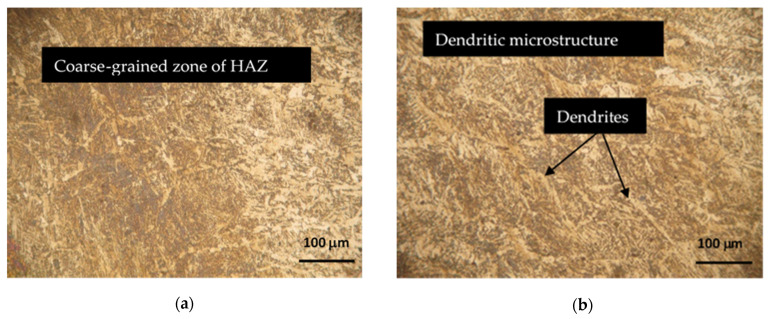
HAZ (left side) and FZ (right side) microstructures of Hardox 450 steel specimen groups 1 and 2: (**a**,**b**) specimen group 1; (**c**,**d**) specimen group 2.

**Figure 8 materials-16-05593-f008:**
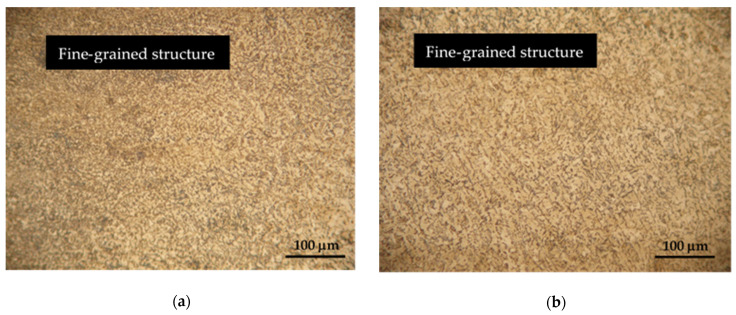
Microstructures of group 3 specimens: (**a**) HAZ microstructure; (**b**) FZ microstructure.

**Figure 9 materials-16-05593-f009:**
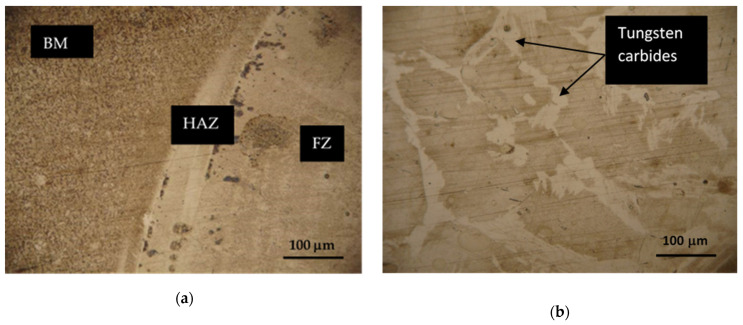
Microstructures of group 4 specimens: (**a**) BM and HAZ microstructures; (**b**) FZ microstructure.

**Figure 10 materials-16-05593-f010:**
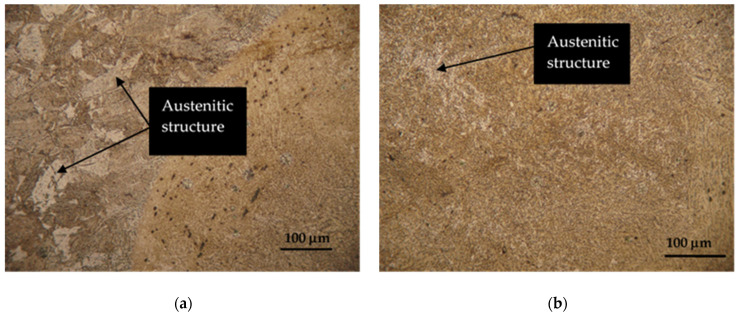
Microstructures of group 5 specimens: (**a**) HAZ microstructure; (**b**) FZ microstructure.

**Figure 11 materials-16-05593-f011:**
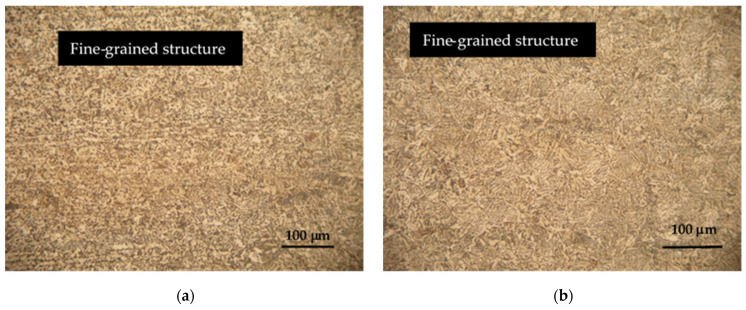
Microstructures of group 6 specimens: (**a**) HAZ microstructure; (**b**) FZ microstructure.

**Figure 12 materials-16-05593-f012:**
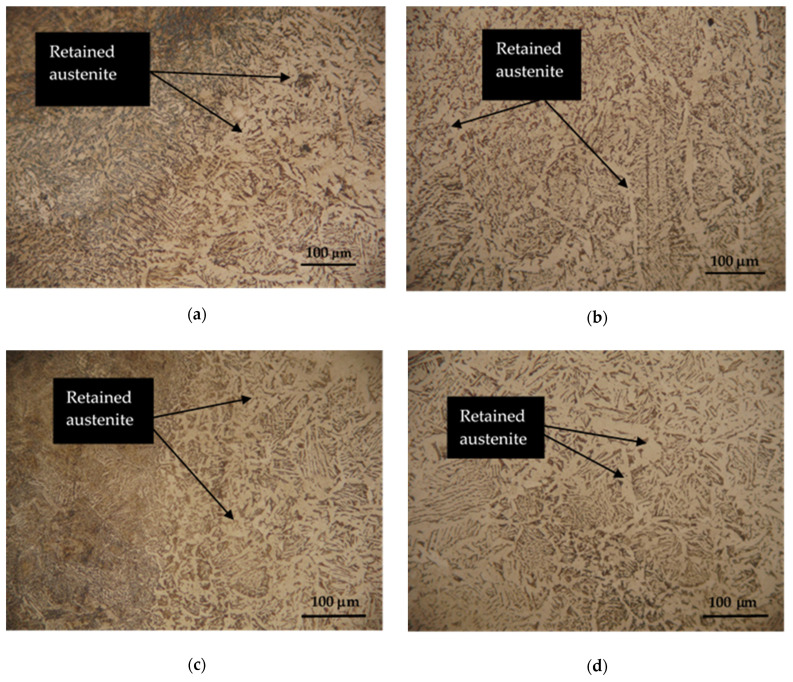
HAZ (left column) and FZ (right column) microstructures of Hardox 450 steel specimen groups 7 and 8: (**a**,**b**) specimen group 7; (**c**,**d**) specimen group 8.

**Figure 13 materials-16-05593-f013:**
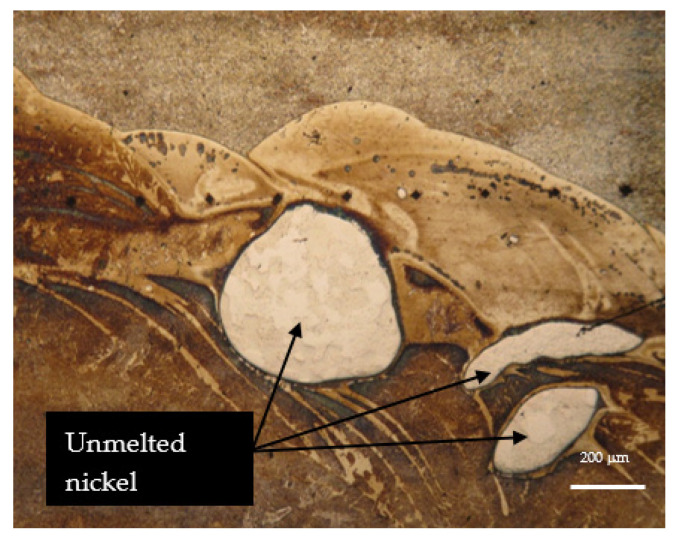
View of the microstructure of the seam with particles of unmelted nickel.

**Figure 14 materials-16-05593-f014:**
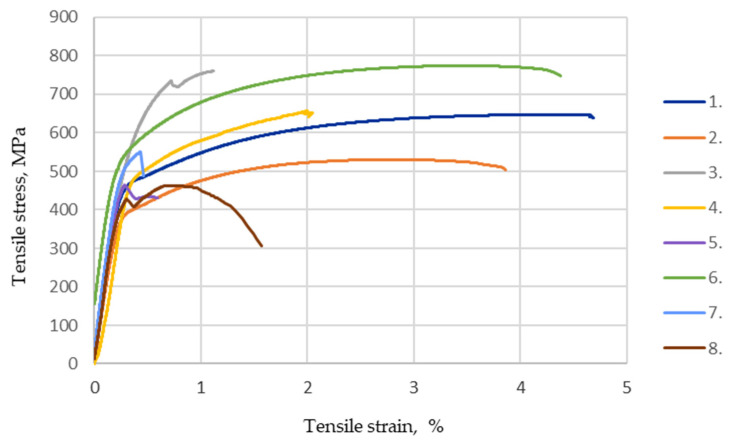
Tensile stress–strain curves of different specimen groups.

**Figure 15 materials-16-05593-f015:**
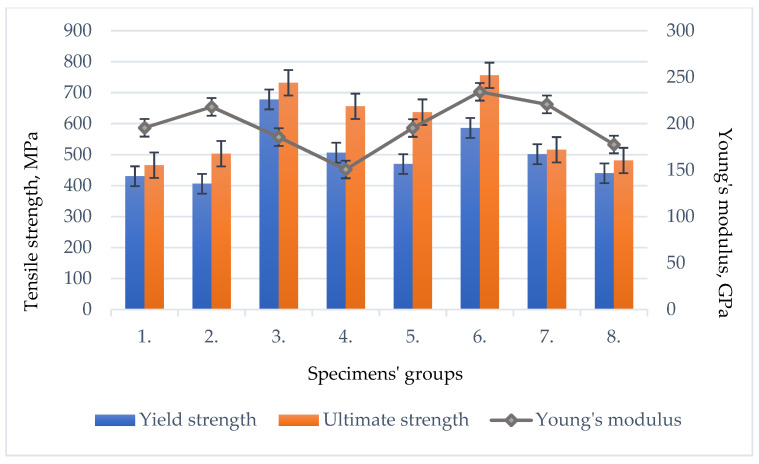
The average values of the yield strength, ultimate strength and Young’s modulus of the tested welds of different groups of specimens.

**Figure 16 materials-16-05593-f016:**
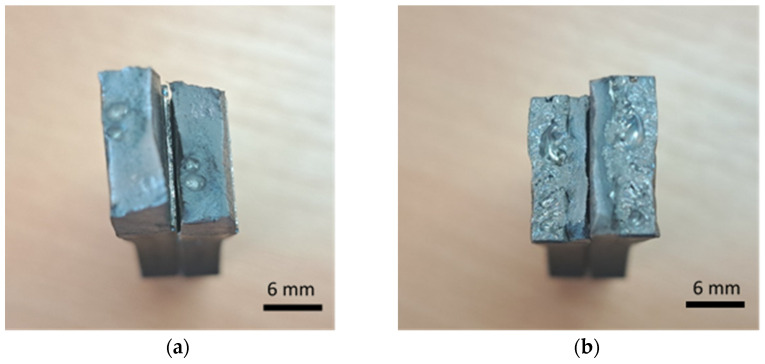
Different types of fracture of welded specimens: (**a**) ductile fracture (specimens of group 5); (**b**) brittle fracture (specimens of group 4).

**Figure 17 materials-16-05593-f017:**
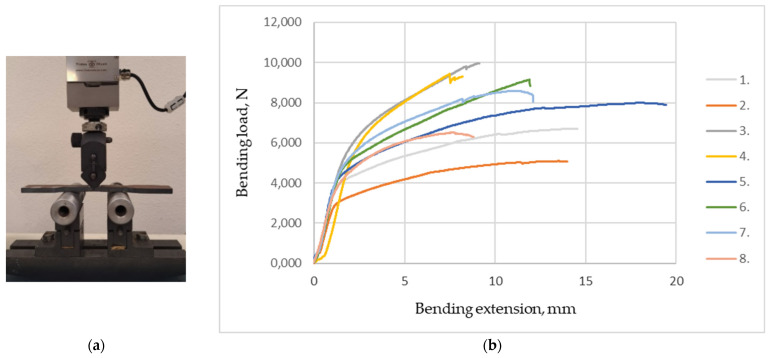
Bend test: (**a**) Experimental setup for bend testing; (**b**) Load—extension curves of the specimen groups.

**Figure 18 materials-16-05593-f018:**
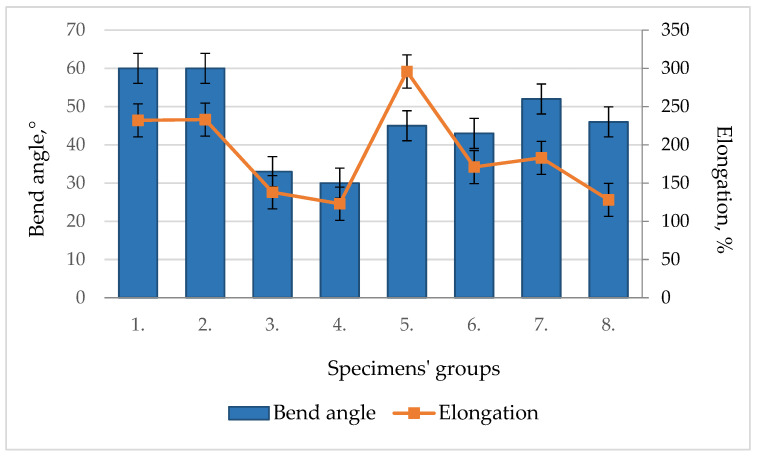
Average values of bend angle and elongation of welded joints.

**Figure 19 materials-16-05593-f019:**
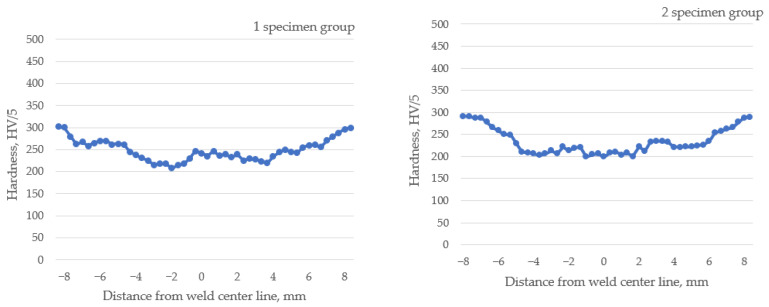
Microhardness profiles of cross sections of specimens (1–8 groups) taken perpendicular to welding direction.

**Table 1 materials-16-05593-t001:** Chemical composition of Hardox 450 steel and filler wire rods.

Grade	Chemical Composition [%]									
	C	Mn	Si	P	S	Cr	Ni	B	Mo	C14
Hardox 450	0.190	1.200	0.250	0.007	0.002	0.050	0.060	0.001	0.024	0.312
ER 70S-6 *	0.150	1.850	1.150	0.025	0.035	0.150	0.150	-	0.150	-
E 7018 *	0.150	1.600	0.750	0.040	0.035	0.200	0.300	-	0.300	-

Fe—balance; * filler rod.

**Table 2 materials-16-05593-t002:** Mechanical properties of Hardox 450 steel and filler wire rods.

Grade	Mechanical Properties				
	Yield Point *R_p_* _(0.2)_ [MPa]	Tensile Strength *R_m_* [MPa]	Elongation *A*_5_ [%]	Impact Strength KCV_-40_ [J/cm^2^]	Brinell Hardness [HBW]
Hardox 450	1200	1400	10	50	425–475
ER 70S-6 *	483	583	28	30–40	-
E 7018 *	400	490	22	27	-

* filler rod.

**Table 3 materials-16-05593-t003:** The main parameters of the welding procedure.

Welding Passes	Current [A]	Arc Voltage [V]	Travel Speed [mm/min]	Flow Rate of Shielding Argon Gas [L/min]	Heat Input [kJ/mm]
1 (root pass)	90	19.5	33.0	12.0	1.92
2 (hot pass)	110	19.5	75.0	10.0	1.03
3 (fill up pass)	110	19.5	75.0	10.0	1.03
4 (cap pass)	110	19.5	75.0	10.0	1.03

Factor of thermal efficiency (TIG) 0.6.

**Table 4 materials-16-05593-t004:** Filler rods and additives used for welding.

Specimen Group	Filler Rod (Standard) ER 70S-6	Filler Rod (Standard) E 7018	Ti Powder	W Powder	NI Powder	Co Powder
1.	✓					
2.		✓				
3.	✓			✓ (50%)		✓ (50%)
4.	✓			✓		
5.					✓	
6.		✓		✓ (50%)		✓ (50%)
7.	✓					✓
8.		✓				✓
9.	✓		✓			

**Table 5 materials-16-05593-t005:** Data of tensile test results of welded joints.

Specimen Groups	Ultimate Tensile Strength, [MPa]	Strain at Ultimate Tensile Strength, [%]	Location of Fracture	Remark/e.g., Fracture Appearance	Hardness, [HV/5]
1.	466.00	0.28	through welding	ductile fracture	252.3
2.	530.46	2.79	through welding	ductile fracture	225.7
3.	760.28	1.12	through welding	brittle fracture	428.1
4.	656.98	2.00	through welding	brittle fracture	478.2
5.	647.19	4.63	through welding	ductile fracture/a neck is formed	344.4
6.	774.39	3.45	through welding	brittle fracture	436.7
7.	551.29	0.43	through welding	brittle fracture	277.8
8.	462.30	0.72	through welding	brittle fracture/negligible gas porosity	349.6

**Table 6 materials-16-05593-t006:** Data of bend test results of welded joints.

Specimen Group	Original Gauge Length, [mm]	Bending Modulus, [GPa]	Bending Yield Strength, [MPa]	Remark/e.g., Fracture Appearance	A View of the Seams after the Bend Tests
1.	6.00	88.12	426.77	Ductile fracture in the tensile zone; open cracks at the edges and center of the seam.	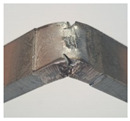
2.	6.00	72.65	329.25	Ductile fracture in the tensile zone; open cracks at the edges of the seam.	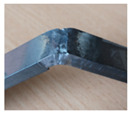
3.	6.02	82.76	491.69	A longitudinal crack is observed in the seam zone.	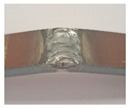
4.	6.01	79.95	532.21	A crack has formed in the tensile zone across the entire width of the seam; small defects are visible.	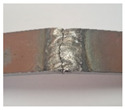
5.	6.00	93.64	464.59	No cracks were detected in the tensile zone; plastic deformations are visible on the sides.	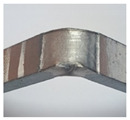
6.	6.00	83.32	506.81	Slight fracture in tensile zone; lateral crack visible in transverse direction.	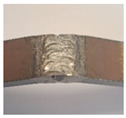
7.	6.01	76.62	464.12	Deep lateral fracture in the tensile zone.	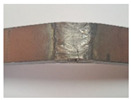
8.	6.01	81.71	391.22	Cracks are observed on both sides of the weld.	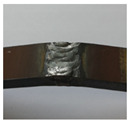

## Data Availability

Not applicable.

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
