# Peer review of "Influence of Additives on the Mechanical Characteristics of Hardox 450 Steel Welds"

_materials, 2023, doi:10.3390/ma16165593_

Round 1
Reviewer 1 Report
This work studies the effects of additives on the mechanical properties of various Hardox 450 Steels. The experiments are conducted with reasonable design, and are comprehensive. Please revise according to the comments below before it gets accepted.
1. Abstract: “The microstructure of the weld zone is definitely related to…” there is no need to say “definitely”.
2. Abstract: “It can be concluded that tungsten, used as an additive, strongly influences hardness of the heat-affected and fusion zones” please be specific - tungsten increases or decreases the hardness? Direct conclusion should be provided here. “Strongly influences” doesn’t provide much information.
3. Should Figure 1 be merged into the experimental section, rather than the introduction part of the manuscript?
4. Conclusion: “Austenite has a face-centered cubic crystal structure that gives the weld high ductility.” I don’t think this is the conclusion from this work. Please remove it.
5. For the hardness measurements - how many indents were made for each sample to provide meaningful statistics?
It's ok.
Author Response
Dear Reviewer 1,
thank you so much for finding the time for analysis of our manuscript and the valuable comments, which are very useful and helped us to revise the manuscript. We have made adjustments according to provided observations.
Response to Reviewer 1 Comments
Point 1. Abstract: “The microstructure of the weld zone is definitely related to…” there is no need to say “definitely”.
Response 1: The suggested word "definitely" is removed from the abstract.
Point 2. Abstract: “It can be concluded that tungsten, used as an additive, strongly influences hardness of the heat-affected and fusion zones” please be specific - tungsten increases or decreases the hardness? Direct conclusion should be provided here. “Strongly influences” doesn’t provide much information.
Response 2: Corrected.
The statement about the effect of tungsten on hardness is more clearly written. Tungsten increases the hardness of the heat-affected and fusion zones. Below is the citation from scientific article that support this claim.
Tiwari et al. [15] (after correction [17]) declared that under the influence of tungsten as an additive (the following is a citation from an article in bibliography) “The transverse tensile strength and the Vickers microhardness of the welded joints were higher by 5% and 65%, respectively than the base material”.
Point 3. Should Figure 1 be merged into the experimental section, rather than the introduction part of the manuscript?
Response 3: Corrected. Figure 1 is moved to the experimental part.
Point 4. Conclusion: “Austenite has a face-centered cubic crystal structure that gives the weld high ductility.” I don’t think this is the conclusion from this work. Please remove it.
Response 4: Corrected. The indicated statement is removed from the conclusions.
Point 5. For the hardness measurements - how many indents were made for each sample to provide meaningful statistics?
Response 5: Explanation. For each specimen, 49 indents were made, starting from the base metal, then the heat-affected zone-fusion zone, and again the base metal at an interval of 0.7 mm, distance of row of indentation from the reference line (surface or fusion line) - 2 mm according to the standard EN ISO 9015-2:2016.
Minor editing of English language required
Response: Additional corrections of technical language of the manuscript were done. Corrections are made using “Track Changes” function.

Reviewer 2 Report
The main and biggest problem is the structure of this manuscript, including:
1)The microstructure should be introduced firstly before the mechanical properties as the microstructure determines the mechanical properties.
2)Introduction was serious lack of logic and the expression is too verbose. Please further refine by better summarize the existing finding and highlight the innovation of this paper.
3) Materials and Experimental Procedure should be further divided into several part based on the experimental aims as 2.1, 2.2 and so on.
4) The description such as“Tensile specimens were prepared perpendicular to the weld seam. The specimens were deformed at a crosshead speed of 10 mm/min until the specimens completely failed.The welded joint of the specimen was not subjected to heat treatment in accordance with the regulations for the welded joint to be tested.”should be introduced in section 2.
5)A part of Discussion should be separated form section 3 Results and Discussion to emphasize the importance and advancement of this manuscript.
In addition, there are several suggestion, as following:
1) The dimensions of specimen should be directly marked on the figure 5.
2) It is a very critical issue whether there was non-destructive inspection before the mechanical experiment to check the welding quality, and whether there is a repeatable experiments to prove the accuracy of the experimental results.
3)Please explain why do some have yield platforms and others do not in figure 6?
Lots of sentences need to be simplified for better understanding.
Author Response
Response to Reviewer 2 Comments
Dear Reviewer 2,
thank you very much for your time, insights and suggestions to improve the quality of the article. We have made the following changes based on your observations:
Point 1: The microstructure should be introduced firstly before the mechanical properties as the microstructure determines the mechanical properties.
Response 1: Corrected. The microstructure of welded specimens is introduced firstly before the mechanical properties.
Point 2: Introduction was serious lack of logic and the expression is too verbose. Please further refine by better summarize the existing finding and highlight the innovation of this paper.
Response 2: Corrected. The introduction has been supplemented/corrected based on your recommendations.
Point 3: Materials and Experimental Procedure should be further divided into several part based on the experimental aims as 2.1, 2.2 and so on.
Response 3: Corrected. Materials and Experimental Procedure was divided into several parts according to your recommendations.
Point 4: The description such as “Tensile specimens were prepared perpendicular to the weld seam. The specimens were deformed at a crosshead speed of 10 mm/min until the specimens completely failed. The welded joint of the specimen was not subjected to heat treatment in accordance with the regulations for the welded joint to be tested.” should be introduced in section 2.
Response 4: Corrected. The descriptions you mentioned are actually more appropriate in section 2. They have been moved to section 2 in accordance with your recommendations.
Point 5: A part of Discussion should be separated form section 3 Results and Discussion to emphasize the importance and advancement of this manuscript.
Response 5: Corrected. Results and Discussion sections are presented separately.
Point 6: The dimensions of specimen should be directly marked on the figure 5.
Response 6: Corrected. All the dimensions of specimen are directly marked on the Figure 5.
Point 7: It is a very critical issue whether there was non-destructive inspection before the mechanical experiment to check the welding quality, and whether there is a repeatable experiments to prove the accuracy of the experimental results.
Response 7: Explanation. A careful visual inspection of all specimens using a microscope was performed prior to mechanical testing. Specimens with observed defects in the welds were rejected and no longer used in the experimental studies. An example could be the specimens welded using titanium powder as an additive (Figure 4, after correction Figure 3), which were no longer used in further studies due to the defects formed.
For tensile tests, 8 specimens were prepared for each group of specimens, for bending tests - 6 specimens from each group, for hardness measurements - 4 specimens from each group, and 49 indents were made in each specimen, starting from the base metal, then the heat-affected zone-fusion zone, and again the base metal at an interval of 0.7 mm, distance of row of indentation from the reference line (surface or fusion line) - 2 mm.
Point 8: Please explain why do some have yield platforms and others do not in figure 6?
Response 8: Explanation. Figure 6 shows Specimens for optical investigation. You are probably asking about the Figure 7 (after correction Figure 14)?
Certain metals and alloys have a well-defined yield point or yield platform in the stress-strain curves. The yield point represents the stress level at which the material undergoes a significant increase in deformation without a proportional increase in stress. During this phase, the material is said to experience "yielding" and plastic deformation, meaning that it deforms permanently without returning to its original shape when the stress is removed. On the other hand, an additive such as nickel powder softens the weld zone, and in this case there is no well-defined yield point but instead show a gradual increase in strain with increasing stress. Such a seam exhibits continuous and more ductile behavior without a distinct yield platform.
Moderate editing of English language required.
Response: Additional corrections of technical language of the manuscript were done.
All corrections are made using “Track Changes” function.

Reviewer 3 Report
This manuscript presents the welding investigation of Hardox 450 steel and the factors that must be considered in order to maintain the required mechanical properties of the weld. The influence of cobalt, nickel, tungsten and titanium additions during welding of Hardox 450 steel on the microstructure and mechanical properties of the fusion and heat-affected zones is studied. The manuscript has done a comprehensive empirical investigation and the title is practical, but the following must be applied before publishing:
The abstract needs significant revision and should be rewritten. The importance of research, the purpose of doing it, and innovation should be explicitly added. Also, the output data should be mentioned quantitatively, and the achievements should be presented in more detail.
Choose more appropriate keywords. Some of them (TIG and structures) are not mentioned at all in the abstract.
The manuscript needs general writing and grammar editing.
The first paragraph of the introduction can be more concise. At the end of the introduction, a suitable summary of the importance of the present issue should be provided.
Use the following resources to deepen the introduction. Effects of post-weld heat treatment on the microstructure and mechanical properties of laser-welded NiTi/304SS joint with Ni filler. Microstructure evolution and deformation behavior during stretching of a compositionally inhomogeneous TWIP-TRIP cantor-like alloy by laser powder deposition. Investigation of welding crack in micro laser welded NiTiNb shape memory alloy and Ti6Al4V alloy dissimilar metals joints.
The second section (Materials and experimental procedure) is very well organized. But it is better to consider the following. For example, the information in Table 4 is vague.
Add a scale bar to Figures 3 and 4.
Explain how to choose welding parameters.
How is the welding quality checked?
How many tensile test samples were prepared for each group? How are the reproducibility of mechanical properties results checked? How accurate was the strain measurement?
Figure 7(a) should be deleted. Figure 7(b) does not have the required quality. Change the horizontal axis to elongation in %.
How is the elastic modulus and yield stress derived? Add an error bar to the results of Figure 8. Add scale bar to Figure 9.
Usually, SEM is used to check the fracture pattern, not macro images like in Figure 9.
Different sections of mechanical properties lack analysis and discussion regarding the results that should be corrected. For example, the following should be clearly discussed:
Why sample 6 has the highest mechanical properties?
What is the cause of the different failure mechanisms for sample 5?
Sample 6 has high elongation despite brittle failure. be explained
No comment.
Author Response
Response to Reviewer 3 Comments
Dear Reviewer 3,
thank you for your time, insights and given comments that will help us to improve the quality of the manuscript.
Point 1: The abstract needs significant revision and should be rewritten. The importance of research, the purpose of doing it, and innovation should be explicitly added. Also, the output data should be mentioned quantitatively, and the achievements should be presented in more detail.
Response 1: Corrected. The abstract has been supplemented/corrected based on your recommendations.
Point 2: Choose more appropriate keywords. Some of them (TIG and structures) are not mentioned at all in the abstract.
Response 2: Corrected. More appropriate keywords were selected for the manuscript.
Point 3: The manuscript needs general writing and grammar editing.
Response 3: Additional corrections of technical language of the manuscript were done.
Point 4: The first paragraph of the introduction can be more concise. At the end of the introduction, a suitable summary of the importance of the present issue should be provided. Use the following resources to deepen the introduction:
Effects of post-weld heat treatment on the microstructure and mechanical properties of laser-welded NiTi/304SS joint with Ni filler.
Microstructure evolution and deformation behavior during stretching of a compositionally inhomogeneous TWIP-TRIP cantor-like alloy by laser powder deposition.
Investigation of welding crack in micro laser welded NiTiNb shape memory alloy and Ti6Al4V alloy dissimilar metals joints.
Response 4: Corrected. The introduction has been supplemented/corrected based on your recommendations.
The publications you mentioned are really valuable and have been used to supplement and improve the quality of the introduction section.
Also, all three publications recommended by you are included in the list of references.
Point 5: The second section (Materials and experimental procedure) is very well organized.
Response 5: Thank you very much for such rating.
Point 6: The second section (Materials and experimental procedure) is very well organized. But it is better to consider the following. For example, the information in Table 4 is vague.
Response 6: Corrected. Additional information on the contents of Table 4 is provided in the manuscript.
Point 7: Add a scale bar to Figures 3 and 4.
Response 7: Corrected. Scale bars are added to Figures 3 and 4 (after corrections Figures 2 and 3 respectively).
Point 8: Explain how to choose welding parameters.
Response 8: Explanation. The choice of welding parameters is determined by a combination of factors that affect the welding process and the quality of the resulting weld, including the type of welding process used, the materials to be welded, the thickness of the material to be welded, the electrode used, the filler materials used, the configuration of the joint, the shielding gas used, and the desired welding quality.
Point 9: How is the welding quality checked?
Response 9: Explanation. A visual inspection using a microscope (“microscopic examination”) as the most fundamental method was used. This included inspecting the weld and the surrounding area for visible defects such as cracks, porosity, incomplete fusion, undercuts and irregular bead shape. Specimens with observed defects in the welds were rejected and no longer used in the experimental studies.
Point 10: How many tensile test samples were prepared for each group? How are the reproducibility of mechanical properties results checked?
How accurate was the strain measurement?
Response 10: Explanation. For tensile tests, 8 specimens were prepared for each group of specimens, for bending tests - 6 specimens from each group, for hardness measurements - 4 specimens from each group, and 49 indents were made in each specimen, starting from the base metal, then the heat-affected zone-fusion zone, and again the base metal at an interval of 0.7 mm, distance of row of indentation from the reference line (surface or fusion line) - 2 mm.
For strain measurements the strain sensor DD1 (Hottinger Baldwin Messtechnik GmbH), with nominal displacement ±2.5 mm and accuracy class 0.1 was used.
Point 11: Figure 7(a) should be deleted. Figure 7(b) does not have the required quality. Change the horizontal axis to elongation in %.
Response 11: Corrected. Figure 7(a) is deleted. The horizontal axis is changed in %.
Point 12: How is the elastic modulus and yield stress derived?
Add an error bar to the results of Figure 8.
Add scale bar to Figure 9.
Response 12: Explanation. Conventional yield stress is set at 0.2 percentage of residual deformation according to standard ISO 6892-1:2019 (Metallic materials – Tensile testing – Part 1: Method of test at room temperature). The value of the elastic modulus E during the transverse tensile test is determined in the part of 10-40% of the yield strength (Rp0.2) also according to the standard ISO 6892-1:2019.
E=(ΔR/Δe) ·100%, (3.13 part of the standard)
where: ΔR is change of stress and Δe is percentage extension/strain.
An error bar has been added to the results in Figure 8 (after corrections in Figure 15).
Scale bar has been added to Figure 9 (after corrections to Figure 16).
Point 13: Usually, SEM is used to check the fracture pattern, not macro images like in Figure 9.
Response 13: We agree with your comment.
Point 14: Why sample 6 has the highest mechanical properties?
Response 14: Explanation. Experimental studies showed that of all the specimen groups, the specimens of group 6 had the highest ultimate tensile strength (774.39 MPa), the highest strain at ultimate tensile strength (3.45%), the hardness of the seam was slightly lower compared to only the specimens of group 4 (436.7 HV versus 478.2 HV), the bending modulus was lower compared to the specimens of group 5 only (83.32 GPa versus 93.64 GPA), the bending yield strength was slightly superior only in specimens of group 4 (532.21 MPa versus 506.81 MPa). In addition, the welding zone of group 6 specimens was characterized by a fine-grained structure. After comparing the experimental results of the specimens of all groups, it can be said that the specimens of group 6 have the best mechanical properties.
Point 15: What is the cause of the different failure mechanisms for sample 5?
Response 15: Explanation. In the group of 5 specimens, nickel was used as an additive during welding. Because of nickel, such a seam generally exhibits ductility due to its ability to plastically deform before breaking.
Slight brittle fracture zones in the weld may have appeared due to the presence of hydrogen in the welding process, where hydrogen atoms diffuse into the material and cause a reduction in ductility.
Point 16: Sample 6 has high elongation despite brittle failure. Be explained.
Response 16: Explanation. In a group of 6 specimens, cobalt and tungsten were used as welding additives. They act as solid solution strengthening steel elements that strengthen the steel matrix and increase its resistance to deformation and at the same time its tensile strength. Both of these additives allowed the formation of a fine-grained structure (Figure 17, after corrections Figure 11), that allows a certain ductile behaviour before failure. Also, the tensile test was performed at a relatively low strain rate (the specimens were deformed at a crosshead speed of 10 mm/min), so under these conditions the material might behave in a ductile manner even if it has brittle tendencies. A slow deformation rates can delay local plastic deformation before eventual failure.
Minor editing of English language required
Response: Additional corrections of technical language of the manuscript were done. Corrections are made using “Track Changes” function.

Round 2
Reviewer 2 Report
It can be accepted now